# Extracellular Vesicle Depletion Protocols of Foetal Bovine Serum Influence Umbilical Cord Mesenchymal Stromal Cell Phenotype, Immunomodulation, and Particle Release

**DOI:** 10.3390/ijms24119242

**Published:** 2023-05-25

**Authors:** Rebecca Davies, Shannen Allen, Claire Mennan, Mark Platt, Karina Wright, Oksana Kehoe

**Affiliations:** 1Centre for Regenerative Medicine Research, School of Medicine (Keele University), RJAH Orthopaedic Hospital, Shropshire SY10 7AG, UK; r.l.davies@keele.ac.uk (R.D.); x2p23@students.keele.ac.uk (S.A.); 2Centre for Regenerative Medicine Research, School of Pharmacy and Bioengineering (Keele University), RJAH Orthopaedic Hospital, Shropshire SY10 7AG, UK; claire.mennan@nhs.net (C.M.); karina.wright1@nhs.net (K.W.); 3Department of Chemistry, Centre for Analytical Science, Loughborough University, Loughborough LE11 3TU, UK; m.platt@lboro.ac.uk

**Keywords:** mesenchymal stromal cells, extracellular vesicles, vesicle depletion protocols, immunomodulation

## Abstract

The immunomodulatory properties of MSCs can be recreated using their extracellular vesicles (EVs). Yet, the true capabilities of the MSC EVs cannot be distinguished from contaminating bovine EVs and protein derived from supplemental foetal bovine serum (FBS). FBS EV depletion protocols can minimise this, but vary in terms of depletion efficiency, which can negatively impact the cell phenotype. We explore the impact of FBS EV depletion strategies, including ultracentrifugation, ultrafiltration, and serum-free, on umbilical cord MSC characteristics. Whilst a greater depletion efficiency, seen in the ultrafiltration and serum-free strategies, did not impact the MSC markers or viability, the MSCs did become more fibroblastic, had slower proliferation, and showed inferior immunomodulatory capabilities. Upon MSC EV enrichment, more particles, with a greater particle/protein ratio, were isolated upon increasing the FBS depletion efficiency, except for serum-free, which showed a decreased particle number. Whilst all conditions showed the presence of EV-associated markers (CD9, CD63, and CD81), serum-free was shown to represent a higher proportion of these markers when normalised by total protein. Thus, we caution MSC EV researchers on the use of highly efficient EV depletion protocols, showing that it can impact the MSC phenotype, including their immunomodulatory properties, and stress the importance of testing in consideration to downstream objectives.

## 1. Introduction

The multipotent capacity of mesenchymal stromal cells (MSCs) has, for some time, been thought to represent an important potential mode of action for this cell-based therapy, but today, great interest is also placed on their ability to modulate the immune system [1,2]. This phenomena can be recreated using MSC conditioned media, suggesting it is a contact-independent process conducted by paracrine signalling [3]. One mechanism through which this is governed is via extracellular vesicles (EVs), which are small, membrane-bound particles released from cells, with the purpose of delivering their lipid, RNA, and protein cargo [4]. Historically thought to be an intermediatory mechanism to deliver their contents for degradation, EVs are now established as a mechanism to convey downstream signalling in recipient cells. In fact, the delivery of MSC EV cargo allows them to recreate the abilities of their parental cells. This includes inhibiting leukocyte proliferation [5,6], biasing an anti-inflammatory phenotype [7,8,9], and improving the prognosis of immune-based disorders, both in animal models [10,11,12] and the clinic [13,14].

The term ‘EV’ is used due to their heterogenous nature which makes it difficult to isolate a single subpopulation due to an overlap in characteristics [15]. Despite this, EVs are predominantly characterised by biogenesis [4]. Exosomes (<200 nm) are formed from the invagination of intraluminal vesicles, a process largely governed by ESCT machinery which, along with accessory proteins, dictates vesicle formation and scission [16]. The resulting multivesicular body can then fuse with the lysosome for degradation, or the plasma membrane for the simultaneous release of exosomes for cell–cell communication. In contrast, endosomes, encompassing microvesicles (<1 µm) and apoptotic bodies (<5 µm), are formed by budding from the plasma membrane. This involves the redistribution of the phospholipid bilayer and remodelling of the cytoskeleton, thereby allowing the vesicle to be shed and later fused with the plasma membrane at the target site [17]. Regardless, EVs offer the potential of a multi-target therapy that faces no risk of rejection, malignant transformation, or change of phenotype upon storage, or injection into an inflammatory environment.

Moving forward, MSC EV therapies will be dependent on our understanding of their desirable characteristics and the mechanisms in which they conduct their therapeutic effects. One obstacle to achieving this is the frequent use of foetal bovine serum (FBS) as a supplement to cell culture media in research and some clinical studies, which contaminates the harvest of EV-conditioned media with bovine EVs and proteins that co-isolate during enrichment [18]. This can make it hard to distinguish the true capabilities of MSC EVs since some effects may be attributed to bovine-derived contamination.

The use of FBS in culture is not a novel problem, and the use of serum-free media was proposed as a solution [19]. However, a lack of FBS can deprive cells of proteins, lipids, and growth factors that promote cell adhesion and prevent cell stress [20]. Additionally, it is possible that serum-free conditions may lead to the study of an EV population which reflects this stressed state. Even commercial alternatives can be challenging in terms of price, undisclosed formulation with potential EV contamination, and the need to perform in-house testing [21,22,23,24].

Alternatively, FBS EVs can be depleted [19]. Here, differential ultracentrifugation (dUC) where FBS is spun at 100,000–120,000× *g* overnight was long considered the ‘gold standard’. However, dUC only achieves partial depletion (~70%) with some FBS EVs persisting in culture, along with their associated protein and RNA [24,25,26]. Other suggestions include ultrafiltration (UF), where the use of commercially available centrifugal filters allow the elution of small molecules but not larger ones, including FBS EVs [27].

In the study presented, we seek to determine how FBS EV depletion strategies (dUC, UF, and serum-free) affect the cellular attributes of umbilical cord MSCs (UC-MSCs), evaluating for the first time the impact of these strategies on their immunomodulatory properties. Upon incubation with these different medias for the final 48 h of media conditioning, we also assessed the resulting harvest following EV enrichment. This work will be imperative in supporting the International Society of EVs’ (ISEVs) guidance to standardise the reporting of EV methodology [15], inform cross-study comparisons regarding EV collection media, and advise MSC EV researchers on their choice of media composition for the purpose of EV harvest.

## 2. Results

### 2.1. Efficiency of FBS EV Depletion Protocols

To indicate the level of contamination that can co-isolate upon EV enrichment, we assessed the residual particle number and protein concentration of dUC and UF depleted FBS after isolation using differential ultracentrifugation. Whilst both methods achieved a significant level of depletion, UF proved to be more efficient than dUC (89.77 ± 4.55% particle depletion in UF, *p* < 0.001, vs. 78.27 ± 4.58% in dUC, *p* < 0.01, Figure 1A), particularly when depleting bovine protein (99.11 ± 0.34% protein depletion in UF, *p* < 0.0001, vs. 70.41 ± 6.68% in dUC, *p* < 0.0001, Figure 1B).

### 2.2. Morphology of UC-MSCs after Incubation with EV-Depleted Medias

Upon determining the efficiency of these EV depletion protocols, we sought to determine their effect on UC-MSCs when using the methods to collect 48 h conditioned media. Here, media composition resulted in morphological changes to the UC-MSCs. Whilst both the FBS and dUC-FBS media showed a fibroblastic morphology, the UF-FBS and serum-free conditions resulted in more bipolar forms compared to the multipolar forms seen in the FBS and dUC-FBS conditions (Figure 2). This was seen in all four biological replicates, regardless of their own variations in morphology.

### 2.3. Phenotype of UC-MSCs after Incubation with EV-Depleted Medias

The population doubling time (PDT) was consistent for all the donors cultivated in the FBS and dUC-FBS media, yet the UF-FBS and serum-free media resulted in significantly slower growth (Figure 3A). Interestingly, the PDT of Donor 2 was less influenced by the reduction in the FBS supplementation (Figure 3B). Despite the slower rate of growth under these conditions, there was no change in the cell viability or the presence of surface markers which define the MSC phenotype [1], regardless of the donor or media composition (Table 1).

### 2.4. Immunomodulatory Properties of UC-MSCs after Incubation with EV-Depleted Medias

After confirming that the UC-MSCs were affected by the media composition for the purpose of EV harvest, we sought to explore whether their immunomodulatory properties were similarly influenced. To explore this, we evaluated their ability to produce IDO protein upon IFN-γ stimulation and their suppression of activated peripheral blood mononuclear cell (PBMC) proliferation [28,29]. Here, all donors displayed no IDO positivity without IFN-γ stimulation. Yet, once stimulated under each of the media compositions for 48 h, there was a significant increase in the IDO presence in the cells, which was significantly higher in the FBS and dUC-FBS media when compared to the UF-FBS and serum-free media (Figure 4A). This seemed to correlate with their ability to suppress PBMC proliferation (Figure 4C), a phenomenon shown to be independent of the influence of slower growth, as assessed by comparing the amount of ATP from UC-MSCs (see Appendix A, Figure A1). Again, there was evidence of donor variation, and Donor 2 was observed to be less influenced by the decreasing FBS supplementation (Figure 4B,D).

### 2.5. UC-MSC EV Enriched Conditioned Media Attributes after Incubation with EV-Depleted Media

Since the cell morphology, proliferation, and immunomodulation were influenced by the media composition for the purpose of EV harvest, we explored the particle content, protein ratio, and presence of EV markers in the UC-MSC conditioned media following EV enrichment. The number of isolated particles did not change, except for when the particles were collected from serum-free conditions which showed a significant reduction (Figure 5A,B, see Appendix A for size distributions, Figure A2). Though, since the media composition was shown to influence the rate of growth and particle release [23], we normalised the data to the cell number. This revealed a trend in which decreasing FBS supplementation increased the number of particles, yet this did not account for the lack of particles in the serum-free conditions (Figure 5C,D). Similarly, the particle to total protein ratios were assessed, revealing a similar trend; however, on account of the increased ‘purity’ of the serum-free conditions, the ratios here increased, but not to the level of the UF-FBS (Figure 5E,F).

This was reflected in the results of the MACSPlex Exosome Kit, which was used to confirm the EV nature of the enriched particles. Whilst all conditions were positive for these markers, the CD9, CD63, and CD81 positive particles were more highly represented within the 6 µg of protein inputted in the lower FBS conditions (Figure 5G). However, caution must be advised upon the interpretation of these results, since CD9 and CD81 positivity was identified in the media only controls, suggesting that FBS EVs may contribute to the positive signal identified for these markers, a phenomena similarly seen in other studies of a similar nature [30] (see Appendix A, Figure A3).

## 3. Discussion

MSC EVs show great promise as a cell-free therapy that would benefit inflammatory disorders. Yet, EV research is stunted by FBS, which leads to false assumptions about MSC EVs due to the co-isolation of bovine EVs and proteins. Hence, the EV community recommends the use of FBS depletion protocols or serum-free media [18]. However, we, and others, have confirmed that these methods of depletion are only partially effective. In accordance with the literature, we found dUC to achieve ~70% depletion of particles and proteins, whilst UF proved to be superior by depleting ~90% of particles and completely eradicating the presence of protein (Figure 1) [24,27,31,32].

It is important to consider that FBS provides essential cell nutrients that support MSCs, and its removal can lead to changes in the cell phenotype and functionality [20]. We confirmed this phenomenon in the UC-MSCs, showing that increasing FBS depletion efficiency caused a change in the cell morphology and proliferation, but not their defining surface markers or viability. This differs from the findings in the adipose MSCs, where lesser serum conditions showed no changes in the morphology and no significant reduction in the proliferation, as well as the reduction in the viability seen in canine MSCs, though here, surface markers were also unchanged [27,33]. However, it is important to acknowledge that no clear conclusions can be made on the effect of the MSCs’ retention of stem properties or potential senescence, which could be an avenue of future research to build upon this work.

Imperatively, MSCs are desirable due to their immunomodulatory capabilities, yet we have shown this to be compromised when FBS is depleted. Whilst we are the first to show this in an EV context, there are similarities to a study investigating the percentage of FBS used to culture bone marrow MSCs, in which a <5% FBS composition significantly reduced IDO activity and PBMC proliferation [34]. Alternatively, studies comparing commercial serum-free media to FBS describe variations in the PBMC proliferation, IDO activity, and inflammatory cytokine profile, dependent on the media tested [35,36,37].

In association with the changes to the cell attributes, the profile of the EV enriched UC-MSC conditioned media similarly differs upon increasing FBS depletion. Whilst this study acknowledges that it is impossible to distinguish the contribution of FBS contaminates in these results, the fact that the particle number does not decrease when utilizing EV-depleted media suggests that it may encourage increased particle release from UC-MSCs. This was also observed in adipose MSCs [27], and supports the hypothesis that FBS, and its associated components, may inhibit EV release [38]. However, this does not account for the significant reduction in particle counts in the serum-free conditions which was observed, even when considering differences in cell growth. Typically, serum-free media are associated with a ‘stressed’ phenotype that increases particle production in neuroblastoma cells [39], but serum-free media were also associated with a high degree of apoptosis not seen in UC-MSCs in the current study. We can only speculate that our cells entered a quiescent state upon the removal of FBS support [40], so this theory and its effect on particle release would need to be explored further.

The common EV markers CD9, CD63, and CD81 were present in all conditions in our study, confirming that EVs were identified. Although under increasing EV-depleted conditions, isolated protein represented a greater degree of particles expressing these three tetraspanins. This is likely an indication of the ‘purity’ of particles collected under the serum-free conditions. However, in acknowledgement that serum-free conditions can alter the EV protein composition [39], it is possible that FBS may inhibit the production of CD9, CD63, and CD81 positive particles. While Western blots, similarly normalised by protein input, show that FBS supplemented media have less CD9 and CD63 in the UC-MSC EVs, a study of HEK293 cells (immortalised embryonic kidney cells) showed that the release of tetraspanin positive EVs are highly correlated with the particle number [38,41].

Clearly, the choice of media used to collect the EVs of interest impacts the cell characteristics and the EVs themselves. We advise caution to the MSC EV community, having shown that EV depletion can affect the UC-MSC phenotype, but more importantly, the immunomodulatory capabilities for which they are studied. The choice of media was also shown to impact the profile of EV enriched conditioned media, yet whether this similarly effects the MSC EVs’ immunomodulatory profile requires further investigation. Nonetheless, we are confident that the work presented here reiterates the importance of detailed reporting of EV methodology, informs cross-study comparison, and stresses the importance of testing cell phenotype in the consideration of downstream objectives.

## 4. Materials and Methods

### 4.1. Umbilical Cord MSCs

Human umbilical cord mesenchymal stromal cells (UC-MSCs) were obtained from previous work conducted by Mennan et al. [42]. Briefly, umbilical cords were sourced from the Robert Jones and Agnes Hunt Orthopaedic Hospital, with informed patient consent (10/H10130/62). Isolation occurred by mincing the tissue before enzymatic digestion with 1 mg/mL collagenase I for 1 h at 37 °C [43]. Hereafter, UC-MSCs were expanded on tissue culture plastic before further expansion using the Quantum hollow-fibre bioreactor cell expansion system (Terumo UK Ltd., Surrey, UK).

For this study, passage 5 UC-MSCs were cultured on plastic in DMEM-F12 (Fisher Scientific UK Ltd., Loughborough, UK), supplemented with 10% foetal bovine serum (FBS, Fisher Scientific UK Ltd., Loughborough, UK) and 1% penicillin-streptomycin (P/S, Fisher Scientific UK Ltd., Loughborough, UK) in a 37 °C incubator with 5% CO_2_. After 48 h, UC-MSCs were washed with PBS (Fisher Scientific UK Ltd., Loughborough, UK) thrice, before exchanging for the four different media compositions (see Table 2).

### 4.2. FBS EV Depletion by Differential Ultracentrifugation (dUC)

FBS (18 mL) was placed into 25P polycarbonate thick-walled centrifuge tubes (Koki Holdings Co., Minato-ku, Japan) in a S50A rotor (k factor = 61) and spun at 120,000× *g* for 18 h using a Hatachu Himac Micro Ultracentrifuge CS150NX (Koki Holdings Co., Minato-ku, Japan) at 4 °C [18]. The supernatant was collected, with the tube angled to leave 1/2 cm of liquid, as to not disturb the visible pellet. Afterwards, the collected supernatant was subjected to 0.2 µm filtration twice and a single 0.1 µm filtration. Now deemed dUC-FBS, the supernatant was then used as a substitute for unprocessed FBS whilst conditioning media for EV harvest.

### 4.3. FBS EV Depletion by Ultrafiltration (UF)

FBS (15 mL) was placed into Amicon ultra-15, 100 kDa filters (Merck Millipore Ltd., Watford, UK) and spun at 3000× *g* for 55 min at 4 °C [27]. The retentate was discarded, whilst the filtrate was collected. Now deemed UF-FBS, the filtrate was then used as a substitute for unprocessed FBS whilst conditioning media for EV harvest.

### 4.4. Light Microscopy

Images to assess cell morphology were taken with a Nikon Eclipse TS100 light microscope using a Leica MC190 HD digital microscope camera using ×10 magnification. Scale bars were calibrated and attached using Fiji software (v.1.53c) [44].

### 4.5. Population Doubling Time (PDT)

Population doubling time (PDT) was calculated between passage 5 and 6 using the following equation:PDT = T ln_2_/ln(X_e_/X_b_),
where T is the incubation time in days, and X_b_ and X_e_ are the cell numbers at the beginning and the end of a passage, respectively [45].

### 4.6. Flow Cytometry for MSC Characterisation

UC-MSCs were incubated in flow buffer (2% bovine serum albumin in PBS) containing F_C_ receptor block (BD Biosciences, Swindon, UK) for 10 min at room temperature. Flow buffer was then added to the appropriate volume (100 µL containing 30,000 UC-MSCs per tube) before incubating with fluorochrome-conjugated antibodies. For surface characterisation, an MSC panel of antibodies was used (CD73, CD90, CD105, CD14, CD19, CD34, CD45, HLA-DR; BD Biosciences, Swindon, UK) and incubated with the cells for 30 min at 4 °C in the dark; the cells were then washed and resuspended in flow buffer for analysis.

For intracellular staining of IDO, after blocking, UC-MSCs were fixed with 2% paraformaldehyde for 20 min at room temperature. Then, UC-MSCs were permeabilised with 0.1% saponin with F_C_ receptor block for 15 min at room temperature, followed by incubation with fluorochrome-conjugated antibody targeting IDO for 30 min, at 4 °C, in the dark. Following this, UC-MSCs were washed with 0.1% saponin and resuspended in flow buffer for analysis.

Viability was assessed by incubating UC-MSCs in 1 µg/mL propidium iodide (PI, Sigma-Aldrich, Dorset, UK) for 1 min and analysing the sample straight after incubation. Here, PI discriminates dead cells due to the cell membrane becoming permeable in death, allowing it to bind to the now accessible DNA. Samples were run on the FACS Canto II cytometer (BD Biosciences, Swindon, UK) and the data were analysed using FlowJo Software (v10.7.1).

### 4.7. PBMC Isolation

PBMCs were isolated from whole blood from healthy volunteers under informed consent, collected in EDTA coated tubes. Blood was processed immediately by diluting 1:1 in PBS + 2% FBS and layering onto Lymphoprep (STEMCell Technologies UK Ltd., Cambridge, UK). This was then centrifuged at 400× *g* for 20 min at room temperature with the instrument break set to ‘off’. The buffy coat layer containing the PBMCs was collected and washed twice. PBMCs were resuspended in 10% DMSO + 90% FBS at a density of 2–3 × 10^6^ and stored in liquid N_2_ until required for subsequent experiments.

### 4.8. PBMC Suppression Assay

The PBMC suppression assay was conducted with slight adaptations as per the method created by Herzig et al. [46]. Briefly, 1.5 × 10^4^ UC-MSCs were seeded in +10% foetal bovine serum (FBS) supplemented media in a 96-well plate in triplicate for each donor and condition for testing. After adhering overnight, the UC-MSCs were washed thrice and the media was replaced with the different media compositions for 48 h (see Table 2). UC-MSCs were washed again before exchanging for RPMI-1640 media (+10% FBS, 1% P/S, 1% NEAA, 1% sodium pyruvate, 50 µM β-metacarpoethanol) containing 1.5 × 10^5^ pooled donor PBMCs per well. PBMCs, with the exception of non-activated controls, were activated using CD3/CD28 beads (TransAct, Merck Millipore Ltd., Dorset, UK) at a ratio of 1:200 (5 µL per 1 mL of media containing 1.5 × 10^6^ cells). Cells were co-cultured for 72 h, after which EDTA was added to a final concentration of 1mM and incubated for 5 min at 37 °C to avoid clumping. Thereafter, 50 µL of non-adherent PBMCs were transferred to a white (wall and bottomed) 96-well plate (Optiplate, Perkin Elmer Ltd., Buckinghamshire, UK), mixed with 50 µL CellTiterGlo (Promega Corperation, Chisworth, UK) and agitated for 2 min. For UC-MSCs, all non-adherent PBMCs were removed, the cells were washed, 75 µL of CellTiter Glo and 75 µL of PBS were then added, agitated for 2 min, and 100 µL was transferred to a white (walled and bottomed) 96-well plate. Both plates were incubated at room temperature for 30 min in the dark and luminescence was read on a FLUOstar Omega microplate reader with an ATP (Sigma-Aldrich Co Ltd., Coventry, UK) standard curve.

### 4.9. Extracellular Vesicle Enrichment by Differential Ultracentrifugation

Twenty millilitres of 48 h conditioned media, per T175 (roughly 3 × 10^6^ cells at harvest, 80–90% confluency, >95% viability), was pre-prepared by a 300× *g* spin for 10 min at 4 °C to remove cells, a 2000× *g* spin for 20 min at 4 °C to remove cell debris and was filtered using a 0.2 µm filter to bias small EVs, <200 nm. This was placed into 25 mL polycarbonate tubes (Beckman Coulter, High Wycombe, UK) for ultracentrifugation using a Type 70 Ti Fixed-Angle Titanium Rotor (k-factor = 44, Beckman Coulter UK Ltd., High Wycombe, UK) at 100,000× *g* for 70 min at 4 °C [19]. Non-visible EV pellets were resuspended in 1 mL 0.2 µm filtered PBS before topping up the tubes for a second ultracentrifugation. After, the resulting pellet was resuspended in 60 µL of filtered PBS, snap frozen in liquid N_2_ and stored at −80 °C for analysis to be completed within 1 month. We submitted all relevant data of our experiment to the EV-TRACK knowledgebase (EV-TRACK ID: EV230009) [47].

### 4.10. Particle Concentration

Particle concentration was achieved via tunable resistive pulse sensing (TRPS) using Izon’s qNano gold with a NP200 nanopore (Izon Science Europe Ltd., Lyon, France). Filtered PBS was used to wet the lower and upper fluid cells before calibrating the nanopore stretch using callipers. A stable amplitude of ~130 nA was achieved using 0.46–0.60 V, 47.57–47.71 mm stretch, and application of 3.0 mbar pressure whilst running samples. This was followed by the running of calibration particles (CP200, 1 × 10^9^–1 × 10^10^) under similar conditions to calibrate samples for comparison. Data were recorded and analysed using the IZON Control Suite (v3.4.2.48).

### 4.11. Protein Concentration

Samples were resuspended in RIPA buffer (150 mM sodium chloride, 50 mM Tris-HCl, 1% NP-40, 0.5% sodium deoxycholate, and 0.1% SDS) to a final concentration of 1× for the purpose of EV lysis which was encouraged by sonication using a Sonomatic Langford sonicator (Agar Scientific, Essex, UK) every 5 min for 30 s over a 15 min incubation on ice. Protein concentration was determined using the microBCA protein assay kit (Fisher Scientific UK Ltd., Loughborough, UK), as per the manufacturer’s guidelines.

### 4.12. Flow Cytometry for Extracellular Vesicle Characterisation

The equivalent of 6 µg protein was inputted into the MACSPlex Exosome kit (Miltenyi Biotec Ltd., Surrey, UK), as per the manufacturer’s recommendations. Briefly, EV preparations were diluted to 120 µL using the MACSPlex buffer before adding MACSPlex Exosome Capture Beads and incubating overnight on an orbital shaker. Samples were washed, and a detection cocktail of CD9, CD63, and CD81 was added and incubated for 1 h at room temperature on an orbital shaker. Samples were washed twice again, but with a 15 min incubation for the second wash using the orbital shaker. Data were acquired on the FACS Canto II cytometer (BD Biosciences) and the data were analysed using FlowJo Software (v10.7.1). An example of how this was gated can be found in Appendix A (Figure A4).

## Figures and Tables

**Figure 1 ijms-24-09242-f001:**
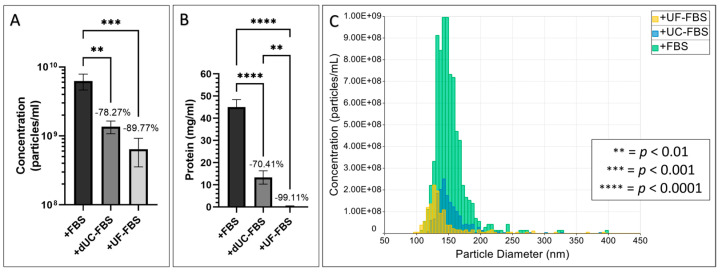
Efficiency of FBS EV depletion protocols via differential ultracentrifugation (dUC) and ultrafiltration (UF). (**A**) Percentage depletion achieved when assessing particle number; (**B**) percentage depletion achieved when assessing protein concentration; and (**C**) representative histograms of particle diameter (nm, *y*-axis) vs. concentration (particles/mL, *x*-axis). Values are expressed as means ± SD (*n* = 3).

**Figure 2 ijms-24-09242-f002:**
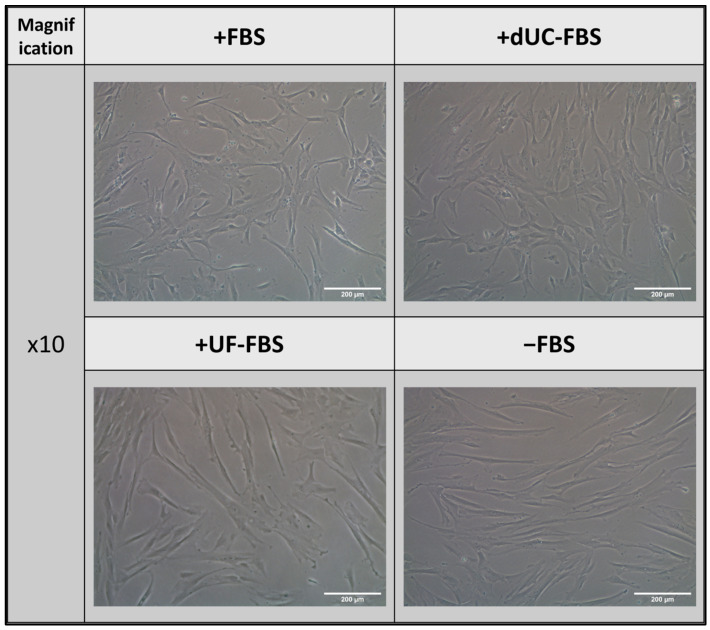
Morphology of UC-MSCs when incubated with different media compositions for the final 48 h of conditioning for the purpose of EV harvest. +FBS indicates media with 10% FBS, +dUC-FBS represents FBS which was depleted by differential ultracentrifugation, +UF-FBS represents FBS which was depleted by ultrafiltration, and −FBS represents base media without FBS supplementation. Images are representative of Donor 3, yet all donors (*n* = 4) displayed a similar change in morphology. Scale bars denote 200 µm.

**Figure 3 ijms-24-09242-f003:**
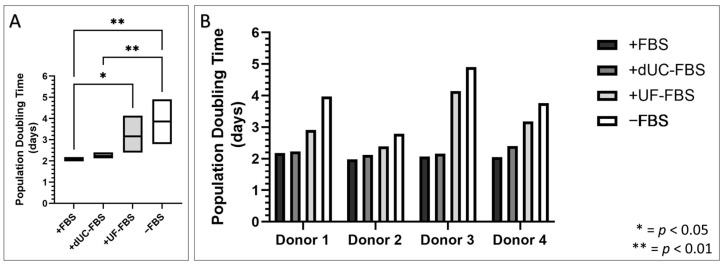
Population doubling time (PDT) of UC-MSCs when incubated with different media compositions for the final 48 h of conditioning for the purpose of EV harvest. +FBS indicates media with 10% FBS, +dUC-FBS represents FBS which was depleted by differential ultracentrifugation, +UF-FBS represents FBS which was depleted by ultrafiltration, and −FBS represents base media without FBS supplementation. (**A**) PDT of biological replicates (*n* = 4) expressed as mean (middle), minimum (bottom), and maximum (top). (**B**) PDT of individual donors placed under the different media compositions.

**Figure 4 ijms-24-09242-f004:**
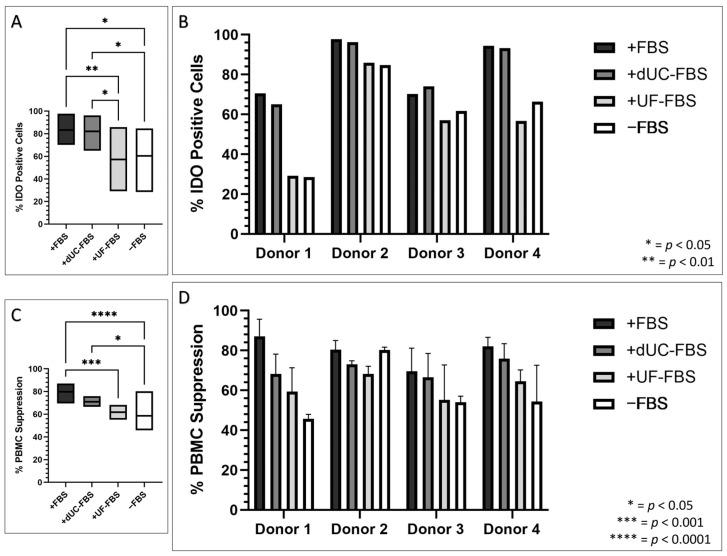
Immunomodulatory properties of UC-MSCs when incubated with different media compositions for the final 48 h of conditioning for the purpose of EV harvest. +FBS indicates media with 10% FBS, +dUC-FBS represents FBS which was depleted by differential ultracentrifugation, +UF-FBS represents FBS which was depleted by ultrafiltration, and −FBS represents base media without FBS supplementation. (**A**) Percentage of IDO positive cells of biological replicates (*n* = 4), when activated with IFN-γ, expressed as mean (middle), minimum (bottom), and maximum (top); (**B**) percentage of IDO positive cells of individual donors when activated with IFN-γ; (**C**) percentage PBMC suppression of biological replicates (*n* = 4) expressed as mean (middle), minimum (bottom), and maximum (top); and (**D**) percentage PBMC suppression of individual donors (*n* = 3, experimental replicates) placed under the different media compositions.

**Figure 5 ijms-24-09242-f005:**
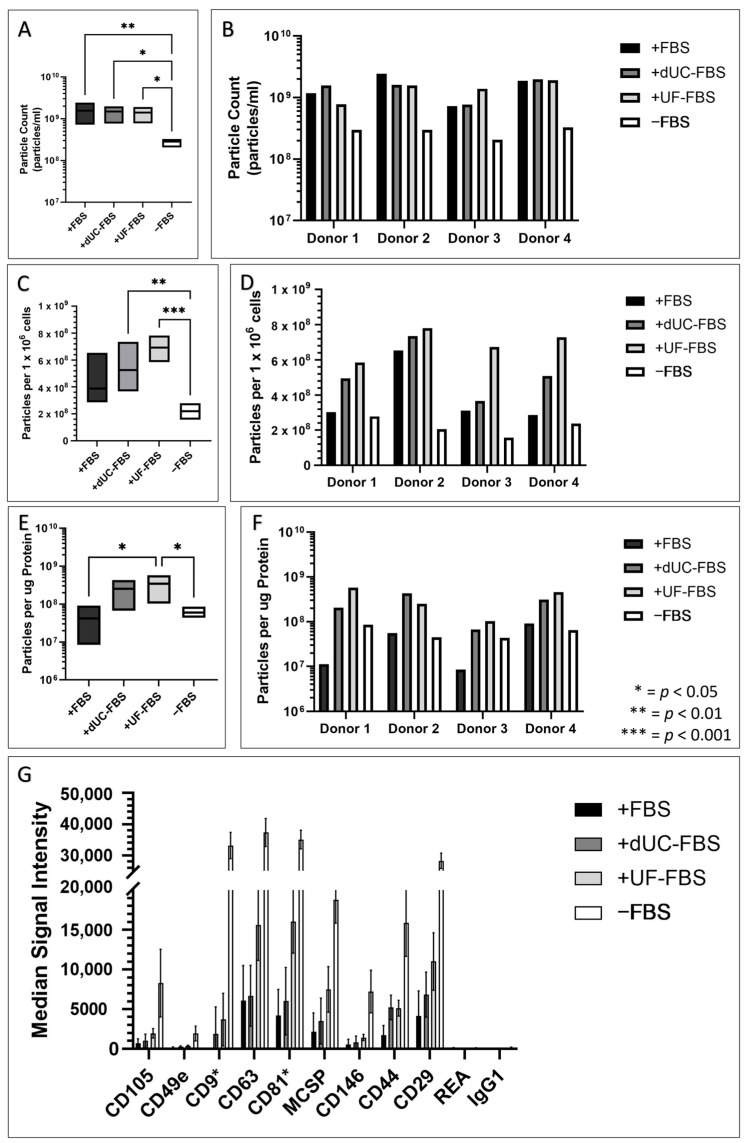
UC-MSC conditioned media profile following EV enrichment when incubated with different media compositions for the final 48 h of conditioning for the purpose of EV harvest. +FBS indicates media with 10% FBS, +dUC-FBS represents FBS which was depleted by differential ultracentrifugation, +UF-FBS represents FBS which was depleted by ultrafiltration, and −FBS represents base media without FBS supplementation. (**A**) Particle counts of biological replicates (*n* = 4) expressed as mean (middle), minimum (bottom), and maximum (top); (**B**) particle counts of individual donors; (**C**) particles per 1 × 10^6^ cells of biological replicates (*n* = 4) expressed as mean (middle), minimum (bottom), and maximum (top); (**D**) particles per 1 × 10^6^ cells of individual donors; (**E**) particles per µg total protein of biological replicates (*n* = 4) expressed as mean (middle), minimum (bottom), and maximum (top); (**F**) particles per µg total protein of individual donors placed under the different media compositions; and (**G**) median fluorescent intensity of positive markers of CD9, CD63, and CD81 positive particles as determined by the Miltenyi MACSPlex Exosome kit, normalised by input of 6 µg of total protein. Asterisks indicate markers whose positivity may be influenced by the presence of FBS EVs which were identified by media only controls.

**Table 1 ijms-24-09242-t001:** Percentage positivity of MSC markers and viable cells after incubation with EV-depleted medias for the final 48 h of conditioning for the purpose of EV harvest, as assessed by flow cytometry. +FBS indicates media with 10% FBS, +dUC-FBS represents FBS which was depleted by differential ultracentrifugation, +UF-FBS represents FBS which was depleted by ultrafiltration, and −FBS represents base media without FBS supplementation. Values are expressed as means ± SD (*n* = 4), and no statistically significant differences were identified.

MSC Marker	+FBS	+dUC-FBS	+UF-FBS	−FBS
CD73	99.85 ± 0.13	99.93 ± 0.15	100.00 ± 0.00	99.93 ± 0.10
CD90	99.55 ± 0.31	99.43 ± 0.78	99.95 ± 0.10	99.63 ± 0.49
CD105	99.68 ± 0.47	99.70 ± 0.48	99.98 ± 0.05	99.78 ± 0.39
CD14	0.81 ± 0.35	0.62 ± 0.31	1.24 ± 0.47	1.20 ± 0.46
CD19	1.40 ± 0.59	1.26 ± 0.37	0.32 ± 0.15	1.23 ± 0.47
CD34	0.33 ± 0.44	1.07 ± 0.73	0.84 ± 0.67	1.44 ± 0.44
CD45	1.18 ± 0.62	1.65 ± 0.34	1.83 ± 0.06	1.38 ± 0.67
HLA-DR	0.18 ± 0.16	0.54 ± 0.31	0.07 ± 0.13	0.33 ± 0.30
Viability	98.45 ± 0.82	98.62 ± 0.80	98.68 ± 0.59	98.57 ± 0.61

**Table 2 ijms-24-09242-t002:** Media compositions explored for the purpose of collecting UC-MSC conditioned media for extracellular vesicle harvest. Each condition includes the associated protocol if FBS was processed, and the final media composition.

Media Condition	Protocol	Composition
+FBS		DMEM-F12 + 10% FBS + 1% P/S
+dUC-FBS	FBS was centrifuged at 120,000× *g* for 18 h at 4 °C. The supernatant was collected, avoiding the last ½ cm of liquid to not disturb the pellet.	DMEM-F12 + 10% dUC-FBS + 1% P/S
+UF-FBS	FBS was loaded onto Amicon ultra-15, 100 kDa filters and spun for 3000× *g* for 55 min at 4 °C. The filtrate was collected.	DMEM-F12 + 10% UF-FBS + 1% P/S
−FBS		DMEM-F12 + 1% P/S

## Data Availability

The datasets generated and/or analysed during the current study are available from the corresponding authors upon reasonable request.

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
