# Peer review of "Extracellular Vesicle Depletion Protocols of Foetal Bovine Serum Influence Umbilical Cord Mesenchymal Stromal Cell Phenotype, Immunomodulation, and Particle Release"

_ijms, 2023, doi:10.3390/ijms24119242_

Round 1
Reviewer 1 Report
The article entitled: "Extracellular vesicle depletion protocols influence umbilical cord mesenchymal stromal cell phenotype, immunomodulation, and particle release" by Davies et al. is very interesting, well designed, well described and the discussion well contestualized.
Good idea to test the 10% of serum obtained from the different ways of separating the EVs and to evaluated the characteristics of MSCs and their EVs.
I have only a suggestion about the title because it is not clear that he "extracellular vesicles depletion protocol" relates to fetal serum, so I suggest to modify the title in
"Extracellular vesicle depletion protocols from fetal serum influence umbilical cord mesenchymal stromal cell phenotype, immunomodulation, and particle release"
In addition, I suggest in table 1 to put the letters indicated the significance where it emerged from the statistical analysis.
Author Response
Thank you for your valued feedback.
- The title of the paper has been changed to read ‘Extracellular vesicle depletion protocols of foetal bovine serum influence umbilical cord mesenchymal stromal cell phenotype, immunomodulation, and particle release’.
- No statistical significances were identified in the MSC marker profile or viability between conditions. To make this clear, I have added the line ‘and no statistically significant differences were identified’ to the figure legend (line 144).
Reviewer 2 Report
Dear Authors,
The manuscript entitled “Extracellular vesicle depletion protocols influence umbilical cord mesenchymal stromal cell phenotype, immunomodulation, and particle release” analyzed the effects of three EV depletion strategies, including ultracentrifugation, ultrafiltration, and serum free, from FBS on umbilical cord MSC characteristics. This manuscript is well-structured and well-written. Interestingly, the data present in this manuscript indicate that EV depletion from FBS using ultrafiltration and serum free media did not impact MSC markers and viability. MSC cultured in media containing EV depleted FBS using ultrafiltration, or serum free media became more fibroblastic, proliferated slower and showed inferior immunomodulatory capabilities. The authors also found that serum free media was shown represent a higher proportion of EV markers when normalized by total protein. This study may be helpful to MSC or MSC-EV based therapy. However, there have already been some studies investigated how to remove EV from FBS more efficiently, and how the EV depleted FBS affect MSC cell culture (e.g. PMID: 29410778). I suggest to reject this manuscript. Please see my comments and suggestions below.
Comments and suggestions
1. Please rephrase line 38-41. EV were not considered to deliver cargos between cells in the beginning. Please check original reference.
2. Line 47-49, please describe EV biogenesis more in detail.
3. Figure 3, in some studies, cells do not proliferate if they are cultured in serum free media. In this study, although cells were cultured in FBS free media, cells still proliferated. Could the authors explain this?
4. How these markers were analyzed? Flow cytometry? Could the authors provide raw data?
5. How the authors can make sure that culture MSC in serum free media did not affect cell viability? Did the authors analyze cell death or apoptosis?
6. Due to the viscosity of FBS, dilution of FBS with media may help to remove FBS EV more efficiently using ultracentrifugation (PMID: 30559953).
7. Flow cytometry plots of the data in figure 5G are needed.
8. Delete lines 391 and 392.
Author Response
The authors would like to thank the reviewer for their careful attention to the presentation and content of our research. We have taken on board the suggestions made and have addressed them with the following changes:
In response to the overall feedback, we are aware of the work done by Kornilov et al., amongst others, and have acknowledged it within this text as the first to suggest ultrafiltration to deplete FBS-EVs. However, it is our belief that this manuscript is of further value to existing literature. First, our work validates that produced by Kornilov et al., and secondly, we add several novel explorations of this method of EV depletion. This includes the effect on MSCs from an umbilical cord source (as opposed to adipose), how it may impact their immunomodulatory capabilities (IDO and PBMC assays, not assessed in any other literature to our knowledge) and resulting particle characteristics (not assessed in other works beyond particle counts). It is our belief these additions would be highly valued in the EV field, especially when so many reviews on this topic query the resulting impact of these growth medias, particularly for MSC-EV therapeutics, as you have pointed out.
- To acknowledge the historical context to EVs, we have added the sentence, ‘Historically thought to be an intermediatory mechanism to deliver their contents for degradation, EVs are now established as a mechanism to convey downstream signalling in recipient cells’ (lines 41-43).
- To expand on EV biogenesis we have added more in-dept information in paragraph 2 of the introduction (lines 51-58): ‘Exosomes (<200nm) are formed from the invagination of intraluminal vesicles, a process largely governed by ESCT machinery which, along with accessory proteins, dictates vesicle formation and scission [16]. The resulting multivesicular body can then fuse with the lysosome for degradation, or the plasma membrane for simultaneous release of exosomes for cell-cell communication. In contrast, endosomes, encompassing microvesicles (<1µm) and apoptotic bodies (<5µm), are formed by budding from the plasma membrane. This involves the redistribution of the phospholipid bilayer and remodelling of the cytoskeleton, thereby allowing the vesicle to be shed and later fuse with the plasma membrane at the target site [17].’
- As indicated in our methodology (lines 275-280), MSCs have been cultured in FBS+ media for 48-hours prior to being washed with PBS three times before changing to the different media conditions for a further 48-hours for the purpose of harvesting EV conditioned media Hence, growth occurs in these first 48-hours, which is accounted for within the population doubling calculations, and any differences are indicated based on what happens within the last 48-hours of conditioning. To make this clear, I have added the following to the figure legends (lines 85-86, 117-118, 133, 140-141, 162 and 194): ‘...for the final 48-hours of conditioning for the purpose of EV harvest’.
- EV markers and viability was assessed by flow cytometry, which is indicated in the methodology (section 4.6.) and has now been indicated in the figure legend (line 140-141). Raw .fcs files can be provided, if necessary, or perhaps made available on request.
- Viability was assessed under each condition using propidium iodide (as stated in methodology, lines 328-333) which can discriminate dead cells due to the cell membrane becoming permeable in death, allowing it to bind to the now accessible DNA. This explanation has now been added to the methodology in lines 330-331.
- Whilst we are aware diluting FBS could improve FBS-EV depletion, we use non-diluted FBS in our methodology, which we hope is clear. This is due to limitations in our EV research set up. However, this is a discrepancy seen throughout the field, due to a wide variation in protocols for ultracentrifugation-based depletion. We hope that the reporting of our depletion efficiencies, as seen in Figure 1, will be informative to our audience. However, it is our understanding that we generate depletion efficiencies like those reported in the literature.
- Representative figures of the data generated in Figure 5G have been added in the appendix as Figure A4.
- Lines 391 and 392 (as in the original submitted manuscript) have been deleted – although these may need to be added back for access of .fcs files.
Reviewer 3 Report
In this research paper, the authors showed FBS EV depletion protocols can minimize negative impact in cell phenotype.
They explored the impact of FBS EV depletion strategies with three different methods: ultracentrifugation, ultrafiltration, and serum-free, on MSC.
They showed that depletion efficiency, seen in ultrafiltration and serum-free, did not impact MSC markers or viability, MSCs did become more fibroblastic, had slower proliferation, and showed inferior immunomodulatory capabilities.
The manuscript contains interesting remarks, and is well-written and organized.
To improve the scientific impact of their manuscript, the authors could consider some comments:
The bibliography could be more updated with recent papers.
Authors should demonstrate that the remaining cells are stem cells too. For this end, assays, such as CFU, differentiation, presence of stemness markers should be evaluated. Even senescence is another important thing that the authors should consider. However, if not able to perform additional experiments, authors could briefly discuss these points.
The manuscript contains interesting remarks, and is well-written and organized. The bibliography mast updated with recent papers.ffwfggr ! "# $% ""# $% &'$%'"'' ' &# '' ! "# $% ""# $% &'$%'"'' ' &# '' ! "# $% ""# $% &'$%'"'' ' &# ''
Author Response
Thank you for your valued feedback.
- We have used the terminology ‘mesenchymal stromal cell’ throughout in acknowledgement of our lack of CFU and differentiation capacity, however, we agree that such limitations should be acknowledged and have added the following in paragraph 2 of the discussion: ‘However, it is important to acknowledge that no clear conclusions can be made on the effect of the MSC’s retention of stem properties or potential senescence which could be an avenue of future research to build upon this work.’ (line 223-225).
- To check our manuscript contains the most up-to-date information, we have double checked the literature using relevant terms for the past 5 years and added any references we believe to be relevant and/or informative, including:
- [30] Pham, C. V.; Midge, S., Barua, H., Zhang, Y., Nguyen, T.N.G., Barrero, R.A., Duan, A., Yin, W., Jiang, G., Hou, Y.; Zhou, S. Bovine extracellular vesicles contaminate human extracellular vesicles produced in cell culture conditioned medium when ‘exosome-depleted serum’ is utilised. Archives of Biochemistry and Biophysics, 2021, 708, 108963.
- [32] Haghighitalab, A.; M Matin, M.; Khakrah, F.; Asoodeh, A.; Bahrami, A.R. Cost-effective Strategies for Depletion of Endogenous Extracellular Vesicles from Fetal Bovine Serum. Journal of Cell and Molecular Research, 2020, 11(2), 42-54.
- [33] Kuwahara, Y.; Yoshizaki, K.; Nishida, H.; Kamishina, H.; Maeda, S.; Takano, K.; Fujita, N.; Nishimura, R.; Jo, J.I.; Tabata, Y.; Akiyoshi, H. Extracellular vesicles derived from canine mesenchymal stromal cells in serum free culture medium have anti-inflammatory effect on microglial cells. Frontiers in Veterinary Science, 2021, 8, 633426.
- [41] Forteza-Genestra, M.A.; Antich-Rosselló, M.; Calvo, J.; Gayà, A.; Monjo, M.; Ramis, J.M. Purity determines the effect of extracellular vesicles derived from mesenchymal stromal cells. Cells, 2020, 9(2), 422.
Round 2
Reviewer 2 Report
Dear Authors,
Thank you for revising the manuscript entitled "Extracellular vesicle depletion protocols influence umbilical cord mesenchymal stromal cell phenotype, immunomodulation, and particle release". The manuscript is well written and well structured. I suggested to accept the manuscript.